# Kinetic Study on Alpha-Form Crystallization of Mixed-Acid Triacylglycerols POP, PPO, and Their Mixture

**DOI:** 10.3390/molecules26010220

**Published:** 2021-01-04

**Authors:** Ken Taguchi, Akihiko Toda, Hironori Hondoh, Satoru Ueno, Kiyotaka Sato

**Affiliations:** 1Graduate School of Advanced Science and Engineering, Hiroshima University, Higashi-Hiroshima 739-8521, Japan; atoda@hiroshima-u.ac.jp; 2School of Food and Nutritional Sciences, University of Shizuoka, Shizuoka 422-8526, Japan; hondoh@u-shizuoka-ken.ac.jp; 3Graduate School of Integrated Sciences for Life, Hiroshima University, Higashi-Hiroshima 739-8528, Japan; sueno@hiroshima-u.ac.jp (S.U.); kyosato@hiroshima-u.ac.jp (K.S.)

**Keywords:** fat, crystallization kinetics, polymorphism, X-ray diffraction, polarized optical microscopy

## Abstract

The crystallization behavior of the metastable α form of triacylglycerols (TAGs) plays a critical role as a precursor for the crystallization of more stable $\beta^{\prime }$ and β forms for various applications in food and pharmaceutical products. However, precise analysis of the crystallization kinetics of α has not been performed, likely due to its rapid and complex behavior. This paper presents the observation results of the initial stages of the isothermal crystallization kinetics of α forms of 1,3-dipalmitoyl-2-oleoyl-glycerol (POP), 1,2-dipalmitoyl-3-oleoyl-*rac*-glycerol (*rac*-PPO), and molecular compound (MC) crystals of a POP/*rac*-PPO (1/1) mixture (MC${}_{\mathrm{POP}/\mathrm{PPO}}$) using synchrotron radiation time-resolved X-ray diffraction and polarized optical microscopy. In all the TAGs, α crystals with a worm-like morphology started to grow rapidly in the first stage. Then, the α crystals slowly transformed into more stable forms in different manners for different TAG samples. In POP, the conversion was simple, as the α-2 form transformed into γ-3, whereas in *rac*-PPO, the lamellar distance values of the α-2 form continuously decreased with time and changed into the α-3 form. In the MC${}_{\mathrm{POP}/\mathrm{PPO}}$ crystals, in contrast, separate crystallization of α-2 of a *rac*-PPO fraction initially occurred, followed by the crystallization of α-2 of POP, and the two α forms merged into α-2 of MC${}_{\mathrm{POP}/\mathrm{PPO}}$. This separate crystallization was caused by large differences in the crystallization kinetics of the α forms of POP and *rac*-PPO.

## 1. Introduction

Triacylglycerol (TAG) crystals possess different polymorphic forms, typically called α, $\beta^{\prime }$, and β, which are defined by unique subcell structures: the hexagonal subcell (H) for α, orthorhombic-perpendicular (O${}_{⊥}$) for $\beta^{\prime }$, and triclinic-parallel (T${}_{\|}$) for β [1]. The three forms differ in their molecular structure, melting and crystallization properties, crystal morphology, and crystal-crystal network, among other properties. The crystallization behavior of TAGs has been studied from the perspective of fundamental aspects and applications in a polymorph-dependent manner [2,3,4]. For example, solid lipid nanoparticles (SLNs) have been studied as carriers of functional substances in pharmaceuticals [5,6,7] and foods [8,9], where the polymorphism of nanostructured lipid crystals is closely related to their functionality, such as loading ability and release control. In particular, the controlled preparation of TAG crystalline particles of the metastable α polymorph is highly effective for the delivery of active drug substances. Therefore, the crystallization and stabilization of the α form in TAG nanoparticles are very important [10,11,12]. In contrast, TAG crystals in emulsion systems, such as butter, margarine, and fat spread, should be in the $\beta^{\prime }$ form, as the crystal morphology of $\beta^{\prime }$ can facilitate the formation of fine crystal networks [13,14,15,16]. Finally, with regard to the most stable polymorph β, the high-density crystal network in chocolate using cocoa butter and related TAGs can be revealed solely by β crystals [17,18,19,20,21]. In addition, oleogels using high-melting TAG crystals are formed by tiny β crystals after the tempering process [22,23].

In contrast to $\beta^{\prime }$ and β, few studies on crystallization behavior have been conducted for the α form. This is likely because α is the least stable polymorph, and therefore, the rapid kinetics of its crystallization and transformation processes have made precise studies difficult. However, recent studies [22,23,24,25] have demonstrated that the α form plays an important role in various applications in pure and mixed TAG systems. For example, an oleogel composed of high-melting TAG β crystals (β-fat gel) is formed by first crystallizing the α crystal and causing a rapid transformation from α to β crystals [22,23]. This implies that β-fat gel cannot be formed when the first crystallized forms are $\beta^{\prime }$ or β and that α crystals may play the role of a precursor form for the crystallization of the β form. Similar processes are involved in the formation of margarine and fat spread, in which the rapid transformation from the first crystallized α form into the $\beta^{\prime }$ form is better controlled during the tempering procedure of rapid cooling and the subsequent heating processes in the crystallizer [24,25].

The significance of the crystallization of α has also been demonstrated in a new type of fat blending using the concept of molecular compound (MC) crystal formation [26]. For example, MC crystals are formed in binary mixtures of 1,3-dipalmitoyl-2-oleoyl-*sn*-glycerol (POP) and 1,3-dioleoyl-2-palmitoyl-*sn*-glycerol (OPO) (MC${}_{\mathrm{POP}/\mathrm{OPO}}$), POP and 1,2-dipalmitoyl-3-*rac*-oleoyl-glycerol (*rac*-PPO) (MC${}_{\mathrm{POP}/\mathrm{PPO}}$), and other mixtures [27,28]. It has recently been discovered that MC${}_{\mathrm{POP}/\mathrm{OPO}}$ crystals exhibit functional properties in edible fat applications [29,30]. It has also been reported that when the mixture of POP and OPO is melted and rapidly cooled, the first crystallized form is α, in which the two TAG components are separated. However, MC crystals are formed in parallel with the polymorphic transformation from α to β during the heating processes [30]. With respect to the SLNs, the α crystals of TAGs produce a higher loading capacity of oil-soluble drug materials than other polymorphic forms [31]. Therefore, the precise study of the structural and kinetic properties of the crystallization of the α form of TAGs is highly significant.

Almost all TAG crystals have α forms as the least stable polymorph [1]. It has been reported [17,21,30,32] that the crystallization behavior of α in saturated-unsaturated mixed-acid TAGs, such as POP, PPO, 1,3-distearoyl-2-oleoyl-glycerol (SOS), and 1,2-distearoyl-3-*rac*-oleoyl-glycerol (SSO), is more complex than that of monosaturated TAGs, such as tristearoyl-glycerol (SSS) [33]. This may be ascribed to the fact that the interactions between unsaturated and saturated fatty acid moieties play a critical role, which may cause the complex occurrence and crystallization of the α form [1]. Because POP and PPO are major TAGs in important natural fat materials, such as palm oil [34], studying the crystallization behavior of α forms is important. In addition, it should be mentioned that fat blending of saturated-unsaturated mixed-acid TAGs has high potential for application in the preparation of healthy edible fats as trans fat alternatives [35,36] and for saturated fat reduction [37].

This paper presents experimental studies on the crystallization kinetics of α forms of POP, *rac*-PPO, and MC${}_{\mathrm{POP}/\mathrm{PPO}}$ using polarized optical microscopy (POM) and synchrotron radiation time-resolved X-ray diffraction (SR-TRXRD). The results confirm that the kinetic behavior of the crystallization of α differs between POP and *rac*-PPO, which is reflected in the complex crystallization behavior of MC${}_{\mathrm{POP}/\mathrm{PPO}}$. Namely, separate crystallization of the first α of *rac*-PPO and the subsequent α of POP occurs, and the two α forms merge into the MC${}_{\mathrm{POP}/\mathrm{PPO}}$ crystals during subsequent stages of crystallization. This is caused by large differences in the crystallization rates of the α forms of POP and *rac*-PPO. The results of the present study are different from previous results on the crystallization behavior of the α form of MC${}_{\mathrm{POP}/\mathrm{PPO}}$ [30]. The different crystallization behavior of the α forms of POP, *rac*-PPO, and MC${}_{\mathrm{POP}/\mathrm{PPO}}$ are discussed by considering glycerol conformations and chain packing of palmitic acid and oleic acid moieties.

## 2. Results and Discussion

In this section, we first demonstrate the time evolution of SR-TRXRD profiles obtained for the isothermal crystallization of POP at 10 ${}^{\circ }$C, *rac*-PPO at 13 ${}^{\circ }$C, and MC${}_{\mathrm{POP}/\mathrm{PPO}}$ at 10 ${}^{\circ }$C. The time evolution of the crystallinity calculated by wide-angle X-ray scattering (WAXS) is also presented. Second, the results of POM observations are presented for each initial stage of the growth of the α form. The growth temperatures are selected as supercooling for each α form, $\Delta T_{\alpha }=(T_{m,\alpha }-T_{x})$, which becomes almost equal at approximately 5.5 ${}^{\circ }$C, as displayed in Table 1.

### 2.1. SR-TRXRD Measurements

Figure 1 presents the SR-TRXRD results for the crystallization of POP at 10 ${}^{\circ }$C after quenching from the melt (see also Appendix A). After a short induction time, the α-2 form with a short spacing (SS) of 0.42 nm and a long spacing (LS) of 4.6 nm started to grow quickly and stopped growing at approximately 90 s. After maintaining a stable state for some time, a transformation started at approximately 300 s from α-2 to a more stable polymorph, γ-3, a triple-chain-length structure with SS (0.47 nm, 0.39 nm) and LS (7.1 nm). These results are in complete agreement with those reported previously [17].

The SR-TRXRD profiles for *rac*-PPO at 13 ${}^{\circ }$C are presented in Figure 2 (see also Appendix A). The WAXS peak with SS = 0.41 nm, which is typical of the α form, started to appear soon after the quenching to 13 ${}^{\circ }$C and increased in intensity until 40 s. However, the SAXS peak profiles exhibited complex behavior. SAXS peaks with LS = 4.8 nm (001 reflection) and 2.4 nm (002 reflection) appeared as quickly as the WAXS peak with SS = 0.41 nm. In addition, a weak peak of LS = 7.8 nm (001 reflection) and a very strong peak of 3.9 nm (002 reflection), which corresponded to the α-3 form, started to appear at approximately 40 s. Interestingly, the first crystallized α-2 form with LS = 4.8 nm gradually changed to the α-2 form with 4.6 nm (001 reflection) and 2.3 nm (002 reflection) until both reflections disappeared around 180 s.

Figure 3 presents the SR-TRXRD results for MC${}_{\mathrm{POP}/\mathrm{PPO}}$ at 10 ${}^{\circ }$C, which differ from those of POP and *rac*-PPO (see also Appendix A). At the beginning of the crystallization, an α-2 form with SS = 0.41 nm and LS = 5.0 nm grew quickly, followed by the growth of another α-2 structure with LS = 4.9 nm, both of which coexisted and stopped growing at approximately 50 s. In the later stage of the crystallization from approximately 150 s, diffraction peaks from a different α-2 form with LS = 4.6 nm and SS = 0.43 nm started to grow slowly, while the peaks from the first α-2 with LS = 5.0 nm and SS = 0.41 nm remained almost constant. These results agree with those previously reported by Minato et al. [32]. Furthermore, at the last stage of crystallization after 600 s, a SAXS peak of 4.2 nm appeared and increased in intensity, which corresponded to the $\beta^{\prime }$-2 form of MC${}_{\mathrm{POP}/\mathrm{PPO}}$ [32].

Figure 4 displays the time evolution of the crystallinity of POP, *rac*-PPO, and MC${}_{\mathrm{POP}/\mathrm{PPO}}$, which was calculated through the decomposition of the WAXS intensities into those of the melt and crystalline peaks (see Appendix A). For POP, the crystallinity increased up to approximately 0.7 at approximately 70 s until the initial growth of the α-2 form stopped and remained almost constant thereafter. The growth of the γ form after 300 s caused a slight decrease in crystallinity, likely because the decomposition process of the WAXS peaks was affected by the peaks of the γ form. The crystallinity of *rac*-PPO also rapidly increased and reached only approximately 0.4 when the initial α-2 structure stopped growing. Subsequently, the crystallinity of *rac*-PPO gradually increased again up to approximately 0.5 during the transformation to the α-3 structure. This indicates that the α-3 structure developed by incorporating the remaining supercooled melt into the initial α-2 crystals at later stages of isothermal crystallization. For MC${}_{\mathrm{POP}/\mathrm{PPO}}$, the initial increase in crystallinity paused with only 0.35 at approximately 60 s when the first α-2 with LS = 5.0 nm and SS = 0.41 nm stopped growing. Subsequently, the crystallinity gradually increased again from approximately 150 s along with the growth of a second α-2 form with LS = 4.6 nm and SS = 0.43 nm. Together with the SR-TRXRD results in Figure 3, this evolution in crystallinity suggests that the first and second α-2 structures corresponded to those of *rac*-PPO and POP, respectively. Namely, a fraction of *rac*-PPO rapidly crystallized separately while the remaining POP fraction crystallized thereafter, and both crystals likely merged to slowly form the α form of MC${}_{\mathrm{POP}/\mathrm{PPO}}$ at the last stage of α-crystallization.

### 2.2. POM Observations

We conducted POM observations to examine the growth mechanisms of the α form and subsequent structural transformations, in particular the reason why *rac*-PPO first crystallized separately to form MC${}_{\mathrm{POP}/\mathrm{PPO}}$ in the POP/PPO binary mixtures. The results of the POM observations for each system are presented in Figure 5, Figure 6 and Figure 7. Needle or worm-like crystals appeared in all samples after several seconds of induction and rapidly grew until finally colliding with one another. The growth and collisions of the worm-like crystals, which were all α from the SR-TRXRD results in Figure 1, Figure 2 and Figure 3, completed within 10 s. The interference colors of the worm-like crystals under a sensitive color plate indicated that the growth advanced along the chain packing direction, as expected. After the completion of the growth and collisions of the worm-like crystals in the early stage, the fine structures slowly changed with increasing brightness (see Appendix A), which corresponded to the structural transitions observed in the initial and later stages of the SR-TRXRD results provided in Figure 1, Figure 2 and Figure 3.

The growth rates *V* of the worm-like α crystals of each sample in the early stage were determined by measuring the time evolution of the length from the center to the tip of each crystal, and some examples are shown in Figure 8. The growth rates determined by the POM measurements are plotted against the crystallization temperatures $T_{x}$ in Figure 9. The growth of the α form of POP was much slower than that of the α forms of *rac*-PPO and MC${}_{\mathrm{POP}/\mathrm{PPO}}$, both of which were comparable or even equivalent by considering the difference in the melting temperature of α, approximately 3 ${}^{\circ }$C, as presented in Table 1. This indicates that the worm-like crystals observed for MC${}_{\mathrm{POP}/\mathrm{PPO}}$ in the early stage corresponded to those of *rac*-PPO, as suggested by the SR-TRXRD results in Figure 3. The large difference in the growth rates of α between *rac*-PPO and POP caused separate crystallization of *rac*-PPO prior to POP before the crystallization of MC${}_{\mathrm{POP}/\mathrm{PPO}}$.

### 2.3. Complex Crystallization Behavior of α Forms

In the present study, the crystallization kinetics of α forms were precisely examined for mixed-acid TAGs, namely POP, *rac*-PPO, and MC${}_{\mathrm{POP}/\mathrm{PPO}}$, using POM and SR-TRXRD. It was discovered that the rapid growth of worm-like α crystals (α-2) with a double-chain-length lamellar structure occurred in the three TAG samples at the initial stage of isothermal crystallization (Figure 1, Figure 2, Figure 3, Figure 5, Figure 6 and Figure 7). However, subsequent crystallization and transformation were very different among the three TAGs, as summarized in Table 1.

Here, the complex kinetic behavior of the crystallization of the α forms is discussed by considering the polymorphic structures of TAG crystals, for which the subcell structure, chain length structure, chain inclination, and glycerol conformation are illustrated in Figure 10 [1]. The hexagonal (H), orthorhombic perpendicular (O${}_{⊥}$), and triclinic-parallel (T${}_{\|}$) subcell structures (Figure 10a) represent the α, $\beta^{\prime }$, and β polymorphs, respectively. In addition, the monoclinic parallel subcell (M${}_{\|}$) has been reported to represent the γ form of POP [17]. The subcell structures were characterized by the following WAXS peaks of SS: single peak of 0.415 nm for H; double peaks of 0.413 nm and 0.38 nm for O${}_{⊥}$; and triple peaks of 0.458 nm, 0.386 nm, and 0.368 nm for T${}_{\|}$ [1]. Fourier transform infrared spectroscopy and Raman spectroscopic studies revealed that the aliphatic chains were in disordered and ordered conformations in α and other more stable forms, respectively [39]. The chain length structure (Figure 10b) comprised a repetitive sequence of leaflets involved in the unit lamella along the long-chain axis, namely two leaflets in the double-chain-length structure and three leaflets in the triple-chain-length structure. The chain inclination is defined as the angle between the long-chain axis and the lamellar interface (Figure 10c). It is considered that the aliphatic chains are arranged normal to the lamellar plane in α, but are inclined for $\beta^{\prime }$ and β [1]. Finally, two types of glycerol conformations were observed in TAG crystals: tuning fork observed in symmetric TAGs, such as POP and OPO, and chair observed in asymmetric TAGs, such as PPO (Figure 10d) [1].

The isothermal crystallization and subsequent transformation of POP were simple: LS = 4.6 nm and SS = 0.42 nm started to appear and continued to grow until the more stable form of γ with LS = 7.1 nm (001 reflection) and 3.5 nm (002 reflection) and SS = 0.47 nm and 0.39 nm started to appear at approximately 300 s. Based on the LS values of the two forms, it is evident that the evolution of the isothermal crystallization of POP proceeded from α-2 (double-chain-length structure) to γ-3 (triple-chain-length structure), as illustrated in Figure 11a.

The results of the SR-TRXRD analysis of *rac*-PPO differed from those of POP. From the beginning to 600 s of crystallization, the WAXS peak of SS = 0.41 nm did not change at all. This signifies that the polymorphism of the crystals occurring during this period of crystallization was α. However, the first occurring α crystal with LS = 4.8 nm changed to another α with LS = 4.6 nm and disappeared at approximately 180 s. In parallel to this, the third α form with LS = 7.8 nm (001 reflection), 3.9 nm (002 reflection), and 2.6 nm (003 reflection) started to appear at approximately 40 s and was maintained until 600 s, indicating that this α form had a triple-chain-length structure. It should be noted that the intensity of the SAXS peak with LS = 4.6 nm started to decrease soon after the SAXS peak with LS = 7.8 nm appeared, and the SAXS peak with LS = 4.6 nm disappeared when the peak with LS = 7.8 nm was at its maximum at approximately 180 s. These changes are illustrated in Figure 11b.

Mizobe et al. studied the crystallization behavior of enantiotropic PPO (R-PPO and S-OPP), reporting that the first crystallized α-2 with LS = 4.95 nm changed to α-2 with LS = 4.05 nm, as illustrated in Figure 11c [40]. They also reported, however, that the 1:1 mixture of R-PPO:S-OPP, that is, synthetic *rac*-PPO, revealed the same crystallization behavior observed in the present study [40]. Therefore, the complex crystallization behavior of *rac*-PPO illustrated in Figure 11b is caused by the racemic mixing of R-PPO and S-OPP. One hypothesis is the contribution of the chair-type glycerol conformation of R-PPO and S-OPP. At the first stage of crystallization, α-2 was formed by allowing palmitic and oleic moieties to coexist in the same leaflet due to the disordered conformation of the aliphatic chains and glycerol groups. However, this structure may not be stable due to the steric hindrance of the chair-type glycerol conformation, and chain–chain separation between the palmitic and oleic moieties may form α-3. Such chain–chain separation may not occur in α of symmetric TAGs of POP and OPO, as illustrated in Figure 11a,d. The details of this mechanism will be investigated in future work.

The crystallization behavior of MC${}_{\mathrm{POP}/\mathrm{PPO}}$ corresponds neither to POP nor to *rac*-PPO as far as its initial stages are concerned. First, no triple-chain-length structure was observed, signifying that the oleoyl and palmitoyl chains coexisted in the same leaflets. Second, the LS value of the first occurring α changed from 5.0 nm to 4.6 nm, where the latter corresponded to α-2 of POP (Figure 11a) and the second α-2 of *rac*-PPO (Figure 11b) and was very similar to α-2 of MC${}_{\mathrm{POP}/\mathrm{OPO}}$ (Figure 11d) [27]. The crystallization of α-2 with LS = 5.0 nm of MC${}_{\mathrm{POP}/\mathrm{PPO}}$ should have been caused by the separate crystallization of *rac*-PPO and POP due to the large difference in the growth rates of α between them, both of which merged to form α-2 with LS = 4.6 nm and/or $\beta^{\prime }$-2 with LS = 4.2 nm of MC${}_{\mathrm{POP}/\mathrm{PPO}}$. The separate crystallization of the α form of MC${}_{\mathrm{POP}/\mathrm{OPO}}$ was also reported by Nakanishi et al. [30], where the α forms of POP and OPO were crystallized separately under rapid cooling (>40 ${}^{\circ }$C min${}^{-1}$) without the formation of MC crystals. The decrease in the LS value from 5.0 to 4.6 nm may have been partly caused by the stabilization of the glycerol conformation in α-2 of MC${}_{\mathrm{POP}/\mathrm{PPO}}$ (Figure 11e), which was nearly identical to that in α-2 of POP and MC${}_{\mathrm{POP}/\mathrm{OPO}}$. The occurrence of LS = 4.2 nm at approximately 800 s corresponded to the transformation to $\beta^{\prime }$-2 with the aliphatic chains inclined against the lamellar plane. The occurrence of SS = 0.43 nm at approximately 200 s was likely due to the slight deformation of the H subcell structure.

## 3. Materials and Methods

Samples of POP and racemic PPO (*rac*-PPO) purified to more than 99% were purchased from Tsukishima Foods Industry Co., Ltd, Tokyo, Japan, and used without further purification. A 1/1 binary mixture of POP and *rac*-PPO was prepared by mixing equal weight samples at room temperature and melting the mixture above 50 ${}^{\circ }$C.

The SR-TRXRD methods were performed at BL40B2 in the synchrotron radiation facility SPring-8 at the Japan Synchrotron Radiation Research Institute (JASRI) in Hyogo, Japan. The samples were placed into copper cells (diameter of 10 mm and thickness of 0.4 mm) with a 3 mm hole covered with Kapton film windows. The energy of the incident X-rays was 12.4 keV (the wavelength was 0.1 nm). X-ray scattering data were collected simultaneously with a complementary metal-oxide-semiconductor (CMOS) flat panel detector [41,42] for WAXS measurements with a camera length of 89.8 mm and a PILATUS 2M semiconductor detector (DECTRIS Ltd. Baden, Switzerland) for SAXS with 1794 mm. The camera lengths were calibrated with the diffraction patterns of cerium dioxide for WAXS and silver behenate for SAXS. Each sample was exposed for 2 s, and the interval after exposure was 1 s.

POM was performed using a Nikon LV 100 microscope (Nikon Co., Tokyo, Japan) with a sensitive color plate under the crossed Nicol condition. The isothermal crystallization kinetics of each sample was recorded by a CMOS camera with EPIX^®^ XCAP software. The temperature program was controlled by a LINKAM THMS-600 hot stage (Linkam Scientific Instruments Ltd., Tadworth, Surrey, U.K.). All samples were melted at 40 ${}^{\circ }$C for 2 min and quenched at 60 ${}^{\circ }$C/min to each isothermal crystallization temperature.

## 4. Conclusions

To summarize, the crystallization kinetic behavior of the α form of *rac*-PPO and MC${}_{\mathrm{POP}/\mathrm{PPO}}$ is very different from that of POP, most likely due to the interactions of palmitic and oleic acid moieties and glycerol conformation, both of which are affected by the racemic mixing processes of R-PPO and S-OPP. In these processes, molecular interactions among the glycerol backbones and acyl chains of palmitic and oleic acid moieties, which are asymmetrically connected to the three glycerol carbon atoms, may occur in quite complex manners to form multiple α forms. This behavior may also be active in successive transformations from α to the most stable form, which is $\beta^{\prime }$ for *rac*-PPO [40], whereas β for MC${}_{\mathrm{POP}/\mathrm{PPO}}$ [27]. Many of these mechanisms have never been clarified and remain for future studies.

## Figures and Tables

**Figure 1 molecules-26-00220-f001:**
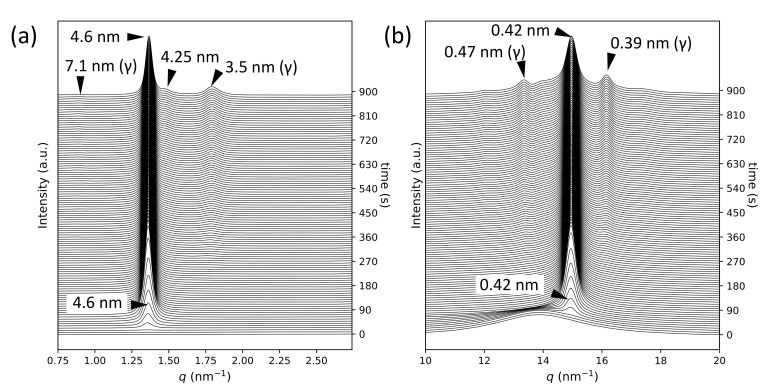
Synchrotron radiation time-resolved X-ray diffraction profiles for POP crystallization at 10 ${}^{\circ }$C: (**a**) small-angle X-ray scattering and (**b**) wide-angle X-ray scattering. The crystallization times are displayed on the right vertical axes.

**Figure 2 molecules-26-00220-f002:**
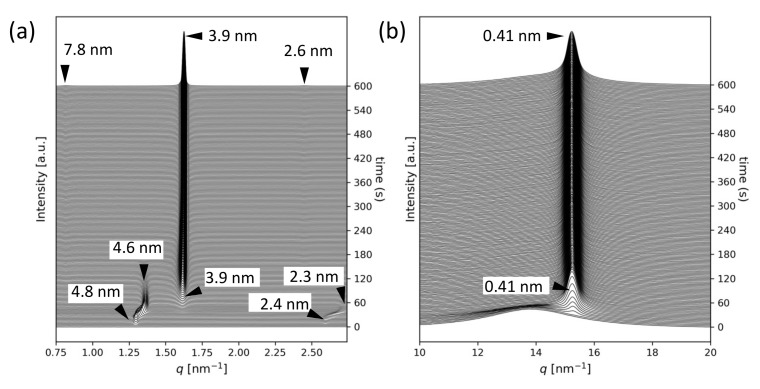
Synchrotron radiation time-resolved X-ray diffraction profiles for *rac*-PPO crystallization at 13 ${}^{\circ }$C: (**a**) small-angle X-ray scattering and (**b**) wide-angle X-ray scattering. The crystallization times are displayed on the right vertical axes.

**Figure 3 molecules-26-00220-f003:**
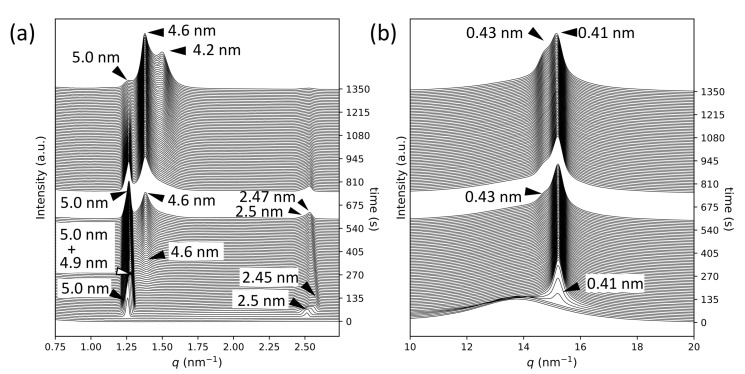
Synchrotron radiation time-resolved X-ray diffraction profiles for MC${}_{\mathrm{POP}/\mathrm{PPO}}$ crystallization at 10 ${}^{\circ }$C: (**a**) small-angle X-ray scattering and (**b**) wide-angle X-ray scattering. The crystallization times are displayed on the right vertical axes.

**Figure 4 molecules-26-00220-f004:**
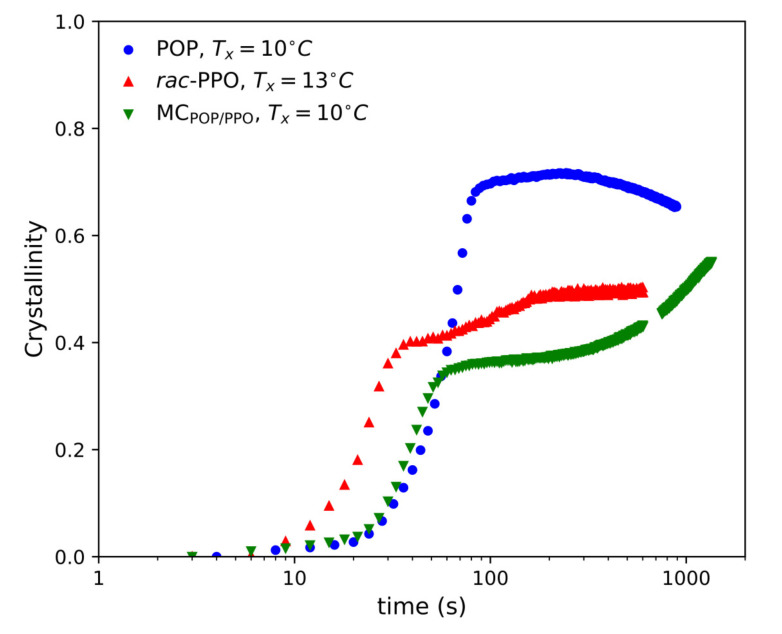
Time evolution of crystallinity during the isothermal crystallization of POP at 10 ${}^{\circ }$C, *rac*-PPO at 13 ${}^{\circ }$C, and MC${}_{\mathrm{POP}/\mathrm{PPO}}$ at 10 ${}^{\circ }$C.

**Figure 5 molecules-26-00220-f005:**
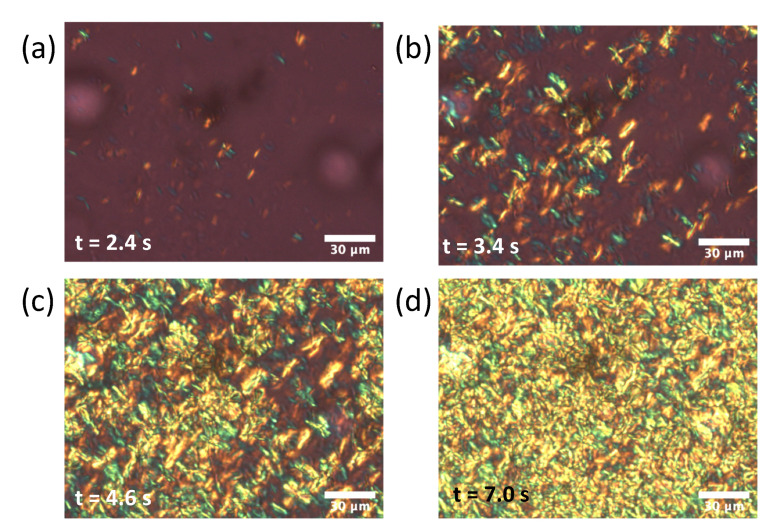
Polarized optical microscopy images of POP with a sensitive color plate during isothermal crystallization at 10 ${}^{\circ }$C. The elapsed time after reaching the crystallization temperature was (**a**) 2.4 s, (**b**) 3.4 s, (**c**) 4.6 s, and (**d**) 7.0 s.

**Figure 6 molecules-26-00220-f006:**
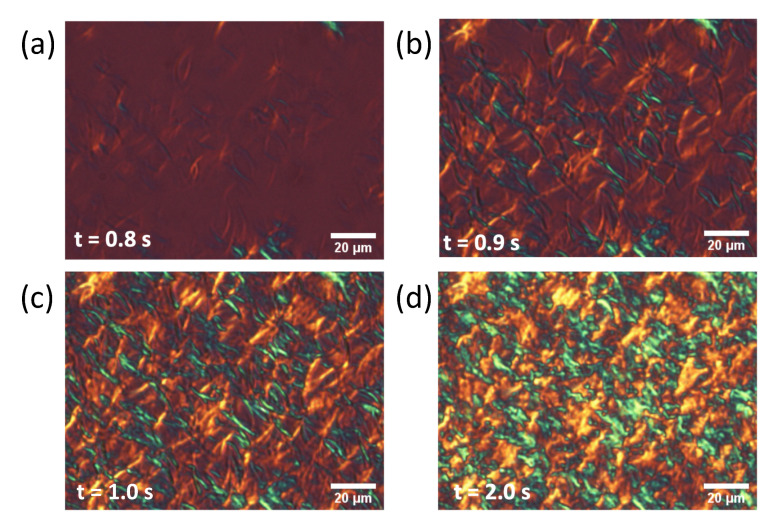
Polarized optical microscopy images of *rac*-PPO with a sensitive color plate during isothermal crystallization at 13 ${}^{\circ }$C. The elapsed time after reaching the crystallization temperature was (**a**) 0.8 s, (**b**) 0.9 s, (**c**) 1.0 s, and (**d**) 2.0 s.

**Figure 7 molecules-26-00220-f007:**
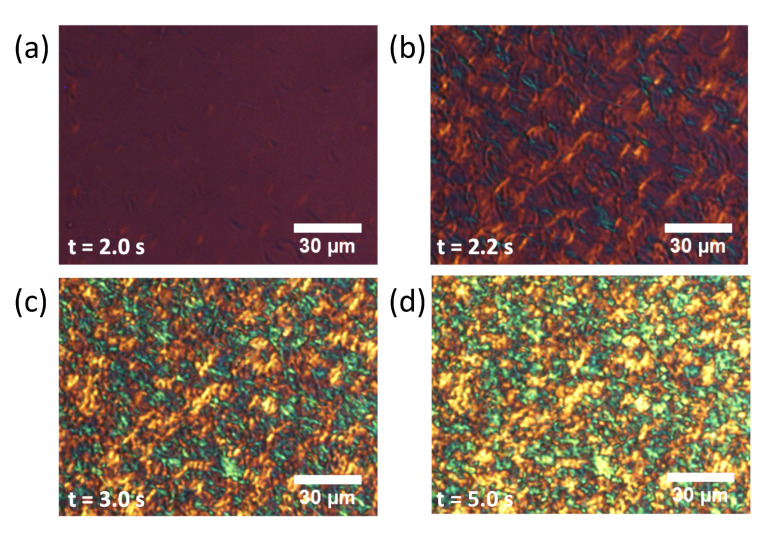
Polarized optical microscopy images of MC${}_{\mathrm{POP}/\mathrm{PPO}}$ with a sensitive color plate during isothermal crystallization at 13 ${}^{\circ }$C. The elapsed time after reaching the crystallization temperature was (**a**) 2.0 s, (**b**) 2.2 s, (**c**) 3.0 s, and (**d**) 5.0 s.

**Figure 8 molecules-26-00220-f008:**
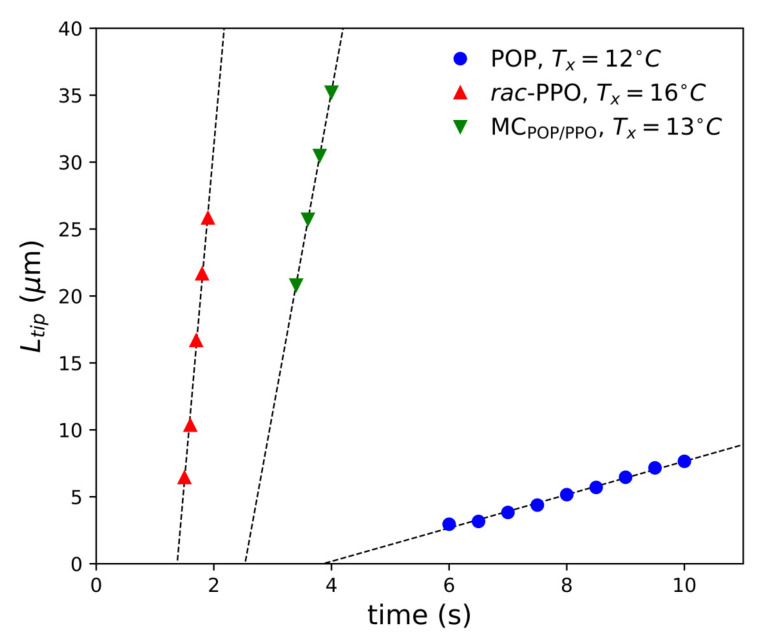
Time evolution of the length from the center to the tip of each worm-like crystal, $L_{tip}$, for POP at 12 ${}^{\circ }$C, *rac*-PPO at 16 ${}^{\circ }$C, and MC${}_{\mathrm{POP}/\mathrm{PPO}}$ at 13 ${}^{\circ }$C. $t_{x}$ represents crystallization time.

**Figure 9 molecules-26-00220-f009:**
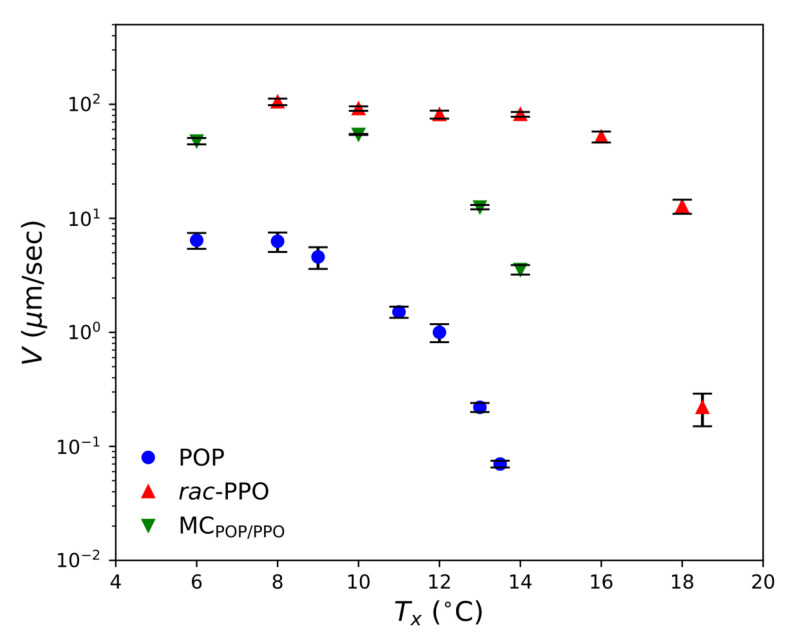
Temperature dependence of the growth rates, *V*, of worm-like α forms for POP, *rac*-PPO, and MC${}_{\mathrm{POP}/\mathrm{PPO}}$. $T_{x}$ represents the isothermal crystallization temperature. Four to six crystals were analyzed for each $T_{x}$, and the mean values are plotted with the standard deviations as the error bar.

**Figure 10 molecules-26-00220-f010:**
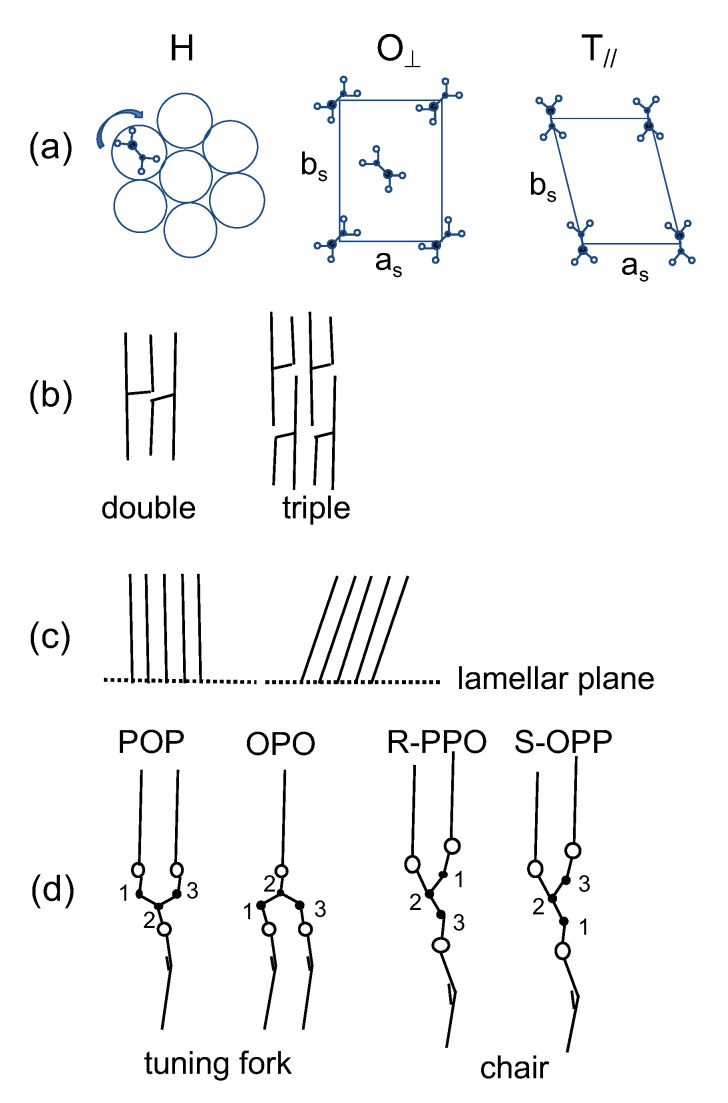
Structure models of polymorphism of triacylglycerols: (**a**) subcells, (**b**) chain length structures, (**c**) chain inclination, and (**d**) molecular shapes of POP, OPO, and enantiotropic PPO.

**Figure 11 molecules-26-00220-f011:**
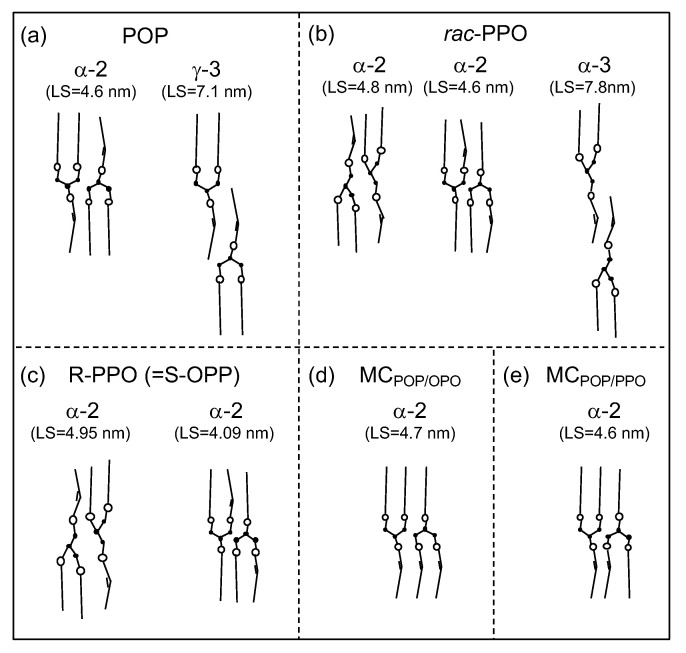
Structure models of metastable α and γ forms of triacylglycerols containing palmitic and oleic fatty acid moieties; (**a**) POP, (**b**) *rac*-PPO, (**c**) enantiotropic PPO, (**d**) molecular compound (MC) of POP and OPO, and (**e**) MC of POP and PPO.

**Table 1 molecules-26-00220-t001:** Melting points of α forms, $T_{m,\alpha }$, of POP [17], *rac*-PPO [38], and MC${}_{\mathrm{POP}/\mathrm{PPO}}$ [32], the crystallization temperatures, $T_{x}$, and the long and short spacing observed for each sample in this study.

	POP	*rac*-PPO	POP/PPO(1/1)
$T_{m,\alpha }$(°C)	15.5	18.5	15.2
$T_{x}$(°C)	10	13	10
long spacing (nm)	4.6	4.8→4.6→3.9	5.0→4.9→4.6
short spacing (nm)	0.42	0.41	0.41→0.41+0.43

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
