# Peer review of "Kinetic Study on Alpha-Form Crystallization of Mixed-Acid Triacylglycerols POP, PPO, and Their Mixture"

_molecules, 2021, doi:10.3390/molecules26010220_

Round 1
Reviewer 1 Report
The paper “Kinetic study on a-form crystallization of mixed-acid triacylglycerols POP, PPO, and their mixture” by Taguchi et al. describes an exhaustive description of the isothermal crystallization of triacylglycerols. The polymorphic characteristics of these substances have particular relevance in the release of pharmaceutical products, and in the beneficial properties of certain foods. There is a great literature detailing the polymorphic forms of TGAs, however, this research shows in detail the mechanisms of forming stable nanostructured lipid crystals, which are of great relevance to the development of products with better functionality.
The transformation of alpha crystals into more stable forms for different TAG and mixtures has been successfully explained by means of polarized optical microscopy, and synchrotron radiation time-resolved X-ray diffraction. The results shown throughout the manuscript are very useful for understanding these crystallization mechanisms. In addition to this, the videos collected in the supplementary information are very didactic.
I recommend the paper for publication as it is.
Author Response
First of all, the authors would like to thank your favorable evaluation and comments for our research and manuscript.
We made minor revisions according to another reviewer's suggestion and believe our manuscript has been improved.
In addition to the revised manuscript, a PDF file that highlights and specifies the revised parts is also included. Please use the PDF file to confirm the revised parts.
Reviewer 2 Report
In this manuscript, the observation results of the initial stages of the isothermal crystallization kinetics of α forms of 1,3-dipalmitoyl-2-oleoyl-glycerol (POP), 1,2-dipalmitoyl-3-oleoyl-rac-glycerol (rac-PPO), and molecular compound (MC) crystals of a POP/rac-PPO (1/1) mixture (MCPOP/PPO) were presented using synchrotron radiation time-resolved X-ray diffraction and polarized optical microscopy. In all the TAGs, α crystals with worm-like morphology started to grow rapidly in the first stage. Then, the α crystals slowly transformed into more stable forms in different manners for different TAG samples. Some problems are listed below.
- Figure 8: The authors should explain how to determine V in Figure 8 from Figures 5-7. Was the length or area of crystal plotted against time to determine V? How many crystals were analyzed to determine V for each condition? A plot of the equivalent length versus time should be plotted to determine V as an example.
- Figure 8: Different symbols should be used for each condition although different colors were adopted. Does the error bar represent the 95% confidence interval of V? It should be clearly specified how many repetitive experiments were performed for each condition.
- Results and Discussion should be combined into “Results and discussion”. “Conclusions“ should be added for the readers.
In summary, this manuscript can be published if the revision is made regarding to the above-mentioned points.
Author Response
First of all, the authors would like to thank the reviewer for detailed and constructive comments and suggestions.
We have carefully reconsidered your comments and believe that our manuscript has significantly improved.
We would appreciate it if you could review this revised manuscript again.
In addition to the revised manuscript, a PDF file "Changes-1049765.pdf" is also included that highlights and specifies the revised parts. Please use the PDF file to confirm the revised parts described below.
The responses for all comments and questions of Reviewer 2 are as follows.
Question & Suggestion 1:
The authors should explain how to determine V in Figure 8.
Reply 1:
We measured the time developments of a length from the center to tip of each needle or worm-like crystal to get the growth rate, V. The explanation is added in line 147-149 of the revised manuscript.
Question & Suggestion 2:
(Figure 8) How many crystals were analyzed to determine V for each condition?
Reply 2:
From four to six crystals were analyzed for each growth temperature to determine V. We added the explanation in the caption of Figure 8 in the revised manuscript.
Question & Suggestion 3:
A plot of the equivalent length versus time should be plotted to determine V as an example.
Reply 3:
We added Figure 9 as an example of some determinations of V, and explained how to determine the growth rates in line 147-149 of the revised manuscript.
Question & Suggestion 4:
(Figure 8) Different symbols should be used for each condition although different colors were adopted.
Reply 4:
We have changed the symbols in Figure 8 and also in Figure 4 for each condition according to the reviewer's suggestion. We also used the same symbols in newly added Figure 9 in the revised manuscript.
Question & Suggestion 5:
(Figure 8) Does the error bar represent the 95% confidence interval of V? It should be clearly specified how many repetitive experiments were performed for each condition.
Reply 5:
We plotted the mean values of V with the standard deviations as the error bar for each condition. We added the explanation about the error bar in the caption of Figure 8.
Question & Suggestion 6:
Results and Discussion should be combined into “Results and discussion”. “Conclusions“ should be added for the readers.
Reply 6:
We have revised the manuscript according to the reviewer suggestion as follows;
1. We have changed the section "Results" into "Results and discussion". (see line 82)
2. The section "3. Discussion" was moved into the last part of "Results and discussion", and the title was revised as 'Complex crystallization behavior of alpha forms". (see line 156)
3. We added the "Conclusion" section at the last part of our manuscript. (see line 247 - 256)
-- Other revisions --
Other revisions are as follows;
Revision 1:
The e-mail address for one author, Kiyotaka Sato, were corrected to "kyosato@hiroshima-u.ac.jp" from "kysato@hiroshima-u.ac.jp".
Revision 2:
"Correspondence: ..." are shown above Abstract on a top page in the revised manuscript.